# Implementation of GA-VirReport, a Web-Based Bioinformatics Toolkit for Post-Entry Quarantine Screening of Virus and Viroids in Plants

**DOI:** 10.3390/v14071480

**Published:** 2022-07-05

**Authors:** Ruvini V. Lelwala, Zacharie LeBlanc, Marie-Emilie A. Gauthier, Candace E. Elliott, Fiona E. Constable, Greg Murphy, Callum Tyle, Adrian Dinsdale, Mark Whattam, Julie Pattemore, Roberto A. Barrero

**Affiliations:** 1eResearch, Research Infrastructure, Academic Division, Queensland University of Technology, Brisbane, QLD 4001, Australia; ruvini.lelwala@aff.gov.au (R.V.L.); z.leblanc@qut.edu.au (Z.L.); marieemilie.gauthier@qut.edu.au (M.-E.A.G.); 2Science and Surveillance Group, Post Entry Quarantine, Department of Agriculture, Fisheries and Forestry, Mickleham, VIC 3064, Australia; candace.elliott@aff.gov.au (C.E.E.); julie.pattemore@aff.gov.au (J.P.); 3Agriculture Victoria, AgriBio, Centre for AgriBioscience, Bundoora, VIC 3083, Australia; fiona.constable@agriculture.vic.gov.au; 4Technology Infrastructure Branch, Information Services Division, Department of Agriculture, Fisheries and Forestry, Canberra, ACT 2601, Australia; greg.murphy@aff.gov.au (G.M.); callum.tyle@aff.gov.au (C.T.); 5Plant Innovation Centre, Post Entry Quarantine, Department of Agriculture, Fisheries and Forestry, Mickleham, VIC 3064, Australia; adrian.dinsdale@aff.gov.au (A.D.); mark.whattam@aff.gov.au (M.W.)

**Keywords:** high-throughput sequencing, small RNA, plant virus and viroid detection, post-entry quarantine, VirReport

## Abstract

High-throughput sequencing (HTS) of host plant small RNA (sRNA) is a popular approach for plant virus and viroid detection. The major bottlenecks for implementing this approach in routine virus screening of plants in quarantine include lack of computational resources and/or expertise in command-line environments and limited availability of curated plant virus and viroid databases. We developed: (1) virus and viroid report web-based bioinformatics workflows on Galaxy Australia called GA-VirReport and GA-VirReport-Stats for detecting viruses and viroids from host plant sRNA extracts and (2) a curated higher plant virus and viroid database (PVirDB). We implemented sRNA sequencing with unique dual indexing on a set of plants with known viruses. Sequencing data were analyzed using GA-VirReport and PVirDB to validate these resources. We detected all known viruses in this pilot study with no cross-sample contamination. We then conducted a large-scale diagnosis of 105 imported plants processed at the post-entry quarantine facility (PEQ), Australia. We detected various pathogens in 14 imported plants and discovered that de novo assembly using 21–22 nt sRNA fraction and the megablast algorithm yielded better sensitivity and specificity. This study reports the successful, large-scale implementation of HTS and a user-friendly bioinformatics workflow for virus and viroid screening of imported plants at the PEQ.

## 1. Introduction

Plant pathogenic viruses and viroids are an increasing threat to global food security [1]. Apart from the economic impact on primary production by reducing yield and quality, plant viruses and viroids can also impair market access and cause severe ecological consequences to native biodiversity. With the ever-growing global movement of plant material and the discovery of novel viruses, improvements in the detection and surveillance of viruses and viroids have become critical to biosecurity and border protection authorities [2,3,4]. Plant quarantine is a biosecurity measure implemented to prevent the entry of economically important plant pests to an area free of those pests or where such pests are present at manageable levels and are officially controlled [5]. Regardless of the point of quarantine enforcement, nationally or regionally, at pre-entry or post-entry, using reliable diagnostic solutions is paramount for making accurate regulatory decisions.

A variety of methods are available to diagnose viruses and viroids in plants. Among these, more conventional methods such as visual inspection and biological assays can result in false positives/negatives due to the absence of disease symptoms, temporal manifestation of symptoms, similarity to abiotic stress phenotypes and symptoms caused by a co-infection of multiple viruses [6]. Moreover, biological assays require a specialist skillset that nowadays is in decline and are resource-intensive, requiring special glasshouse facilities to maintain quarantined plants over long periods (up to 2 years). These challenges add to the cost of biosecurity clearance and delay the release of new genetic material, impacting the ability of Australia’s agricultural sector to maintain its competitiveness [3]. While electron microscopy detection of viruses using particle morphology enables the discovery of infecting viruses, narrowing it down to the species taxonomic level and its low intrinsic sensitivity remain a challenge [7]. Serological and molecular methods are sensitive and specific, allowing pathogen identification up to genus and species levels, but their application focuses on known and well-characterized pests [8]. Furthermore, both serological and molecular assays usually target a single or few closely related pathogens, and they may not readily detect genetically divergent viruses, leading to false-negative detections. Additionally, the increase in the number of viruses of biosecurity concern over time requires implementing a large set of molecular assays, which is increasingly becoming an unsustainable approach to large-scale quarantine testing [3,7].

The advent of high-throughput sequencing (HTS) has revolutionized plant virus diagnostics by facilitating the detection and identification of pathogens without a priori knowledge of infectious status [8]. The large number of data generated through HTS enables detection of all viral agents present, and classification into the lowest taxonomy level is feasible [4,9]. Therefore, HTS can reduce the time and resources required for quarantine screening by eliminating the need to test for a large panel of individual pathogens over a lengthy quarantine period [3,10]. HTS can assist in optimizing the allocation of diagnostic resources during PEQ testing, effectively informing what molecular assays to run to verify HTS-detected pests. Moreover, HTS sequencing can lead to the discovery of new viruses or divergent strains that can be emerging pests [11,12]. With rapid advancements in sequencing technologies and the plummeting cost per base, HTS is becoming an increasingly popular and economical tool for high-throughput virus and viroid diagnostics [9].

Analysis of RNA sequences from host plants is an effective virus and viroid detection method considering most plant viruses have RNA as their genetic material and DNA viruses produce RNA transcripts [12]. Total RNA (commonly ribosomal-depleted), double-stranded RNA and small RNA (sRNA) have been explored as the targeted nucleic acids to be sequenced [13]. The sRNA method explicitly targets the products of the host antiviral response mediated by plant Dicer-like enzymes (DCLs) that cleave viral RNA and replication intermediates into 21–24 nt-long fragments (reviewed in [14]). Moreover, the reduced costs of sRNA-seq and its ability to flag actively replicating viral pathogens compared to RNA-seq have made it a widely adopted approach for viral pathogen diagnostics [10,15,16,17].

Bioinformatics analysis for virus and viroid diagnostics using sRNA-seq data include: (1) quality processing of reads, (2) identifying and removing host reads, (3) de novo and/or reference-guided genome assembly, (4) sequence similarity searches of the contigs against public or custom viral databases and (5) the summarizing and curation of the results. Although theoretically simple, these analyses are often a complex task requiring specialist knowledge in bioinformatics [17]. Several bioinformatics toolkits were recently developed for streamlining the detection of viruses and viroids from host plant sRNA-seq data. Some of these tools include the viral surveillance and diagnosis (VSD) toolkit [10], VirusDetect [18], VirFind [19] and VirReport [20]. In plant quarantine testing settings, the bottleneck for using some of the existing automated toolkits is the requirement for either local installation or knowledge of the Linux operating system and IT security barriers relating to the potentially sensitive nature of the data being generated. Although the tools VirFind [19] and VirusDetect [18] provide public web servers, the individual steps remain hidden from the users. Commercial software such as CLC Genomics Workbench [21] or Geneious Prime [22] enable the building of all-in-one customized workflows with visualization tools. However, these tools incur ongoing licensing fees and require appropriate computational resources at the user’s end to analyze large datasets. As Jones et al. [12] emphasized, the availability of a simple web-based analytical environment is of better use to the research community and diagnosticians because the users can modify processing parameters.

Another challenge associated with the bioinformatics analysis of sequencing data for diagnostic purposes, in general, is the dependency on sequence databases for taxonomic assignment of detections [12,17]. Although public databases such as the GenBank database maintained at National Center for Biotechnology Information (NCBI) are the most up to date, the enormous nature of these databases limits their portability across different computing resources. Moreover, sequencing data submitted by users to these public databases may not be accurate, and annotations provided in the GenBank record, such as the taxonomy assignment, which is crucial for accurate diagnosis, may be inaccurate and/or out of date [23]. Additionally, the descriptors of the sequences in the public databases are not harmonized and lack taxonomic information, posing an additional challenge to validating sequence-homology-based pathogen detections.

Here, we introduce a web-based counterpart of VirReport [20] that is executable on the Galaxy Australia (GA) platform [23], which we called GA-VirReport, and a harmonized plant virus and viroid database (PVirDB). A secondary GA-VirReport-Stats pipeline generates confidence statistics for detections and includes reference-guided consensus sequence generation. This work describes the validation and subsequent implementation of the above resources in screening for viruses and viroids in plants quarantined at the PEQ.

## 2. Materials and Methods

### 2.1. Plant Material

#### 2.1.1. Control Plants

A subset of eight plants from the Gauthier et al. [20] study were selected for comparative pilot implementation and testing of the GA-VirReport workflow. These plants belonging to *Citrus* (*n* = 3), *Prunus* (*n* = 1), *Rubus* (*n* = 1), *Miscanthus* (*n* = 1), *Iris* (*n* = 1) and *Ipomoea* (*n* = 1) genera were maintained in the PEQ glasshouses, Mickleham, Australia, and were re-sampled in March 2021 for the current study.

#### 2.1.2. Imported Plants

A total of 105 imported plants held in quarantine at PEQ were selected for a large-scale study using HTS and GA-VirReport for quarantine testing (Appendix A). These plants were from a range of plant genera, namely *Solanum* (*n* = 9), *Fragaria* (*n* = 14), *Vitis* (*n* = 4), *Malus* (*n* = 3), *Pyrus* (*n* = 6) and *Prunus* (*n* = 69). All plants were sampled between September and November 2021 according to [20].

### 2.2. Nucleic Acid Extraction

For all except *Rubus* and *Fragaria*, 50 mg of the leaf/midrib was subsampled for nucleic acid extraction. For *Rubus* and *Fragaria*, 50 mg of the petioles were used for nucleic acid isolation as per PEQ sampling procedures. Total RNA extractions were performed as previously described [20]. The quality and concentration of total RNA were measured using a NanoDrop™ One/OneC Microvolume UV-Vis Spectrophotometer (ThermoFisher Scientific, Waltham, MA, USA) following manufacturer’s protocols.

### 2.3. Library Construction and Sequencing

Aliquots of 14–15 µL of the total RNA extract were shipped on dry ice to the Ramaciotti Centre for Genomics, University of New South Wales, Australia. The concentration and the purity of the samples were checked using the Epoch Microplate Spectrophotometer (Agilent, Santa Clara, CA, USA), and the RNA integrity was evaluated using the 4200 TapeStation System (Agilent) using the RNA Screen Tape device (Agilent). A total of 100 ng of the quality-checked RNA was used in the library construction using the QIAseq miRNA Library Kit (Qiagen, Hilden, Germany). QIAseq miRNA UDI 96 index plate with unique dual-index adapters was used in the library preparation according to the manufacturer’s protocol. With each library prep batch, Qiagen XpressRef Universal Total RNA (Qiagen) and nuclease-free water were used as the positive and negative controls, respectively. Invitrogen™ Quant-iT™ PicoGreen™ dsDNA Assay Kit (ThermoFisher Scientific) and the LabChip DNA High Sensitivity Reagent kit (PerkinElmer, Waltham, MA, USA) were used to perform quality checks on individual sequencing libraries. Libraries were then normalized, equimolarly pooled and cleaned up with Qiagen QMN beads (Qiagen). Quality check on the cleaned pooled libraries was performed using the Qubit dsDNA HS assay kit on a Qubit Fluorometer (ThermoFisher Scientific) and a TapeStation System (Agilent) High Sensitivity D1000 ScreenTape device (Agilent). Sequencing was performed on the Illumina NovaSeq 6000 instrument with SP 100c flowcell for the control samples and the S1 100c or S2 100c flowcell for the imported samples, generating 75 bp single-end reads with unique molecular identifiers (UMIs). PhiX control (Illumina, San Diego, CA, USA) was used as a control for sequencing runs.

### 2.4. Bioinformatics Analyses

The sequencing provider performed demultiplexing of the raw reads using the bcl2fastq software [24]. All raw FASTQ files from the control dataset were deposited in the Short Read Archive (SRA) database under the BioProject PRJNA752836. We deposited the GA-VirReport workflow for multiple input files (https://doi.org/10.5281/zenodo.6387492) and GA-VirReport-Stats workflow (https://doi.org/10.5281/zenodo.6387504) (Figure 1) at Zenodo. We used these workflows to process raw sequencing data and custom bash scripts for the downstream bioinformatic processing. GA tools used in individual tasks are detailed below.

#### 2.4.1. Preparation of Input Files for the GA-VirReport Workflow

The upload of raw files into the GA platform [23] was carried out via the ‘Upload Data’ tool in the GA toolshed [25] or remotely via an API gateway (unpublish). For each sample, the demultiplexed fastq files were uploaded into individual histories in GA, and a ‘Dataset list’ was created listing all the sequencing files associated with a sample. The multiFASTA file for the PVirDB database (see below) was also uploaded using the ‘Upload Data’ tool. A dataset named ‘non-informative database’ consisting of rRNA, mitochondria and chloroplast sequences derived from plant species held at PEQ was also uploaded to the same history and made publicly available at Zenodo’s data repository (https://doi.org/10.5281/zenodo.6387507). The GA-VirReport workflow for multiple input files was selected from the ‘Workflows’ tab in the GA interface and linked to the respective raw fastq inputs before running the workflow for each sample.

#### 2.4.2. GA-VirReport Workflow

Multiple sequencing read files in the dataset list corresponding to the same sample were merged using the ‘Concatenate multiple datasets tail-to-head (Galaxy Version 0.2)’ tool. Sequencing adapters were then trimmed from reads using the ‘Cutadapt (Remove adapter sequences from FASTQ/FASTA (Galaxy Version 3.5+galaxy1))’ tool, which uses cutadapt (Version 3.5) [26]. The adapters specified were the Illumina TruSeq dual-indexing adapters 5′-AGATCGGAAGAGCACACGTCTGAACTCCAGTCA-3′ as the 3′ adapter and ‘3′-ACACTCTTTCCCTACACGACGCTCTTCCGATCT-5′ as the 5′ adapter. Adapter filtered reads were filtered for UMIs using the ‘UMI-tools extract (Extract UMI from fastq files (Galaxy Version 1.1.2+galaxy2))’ tool, which utilizes UMI-tools (Version 1.1.2) [27]. UMI-extracted reads were then subjected to quality filtering with a Q score threshold of 30 and an unknown base limit of zero using ‘fastp (fast all-in-one pre-processing for FASTQ files (Galaxy Version 0.20.1+galaxy0))’ tool with fastp (Version 0.20.1) [28]. Fastp and ‘FastQC (Read Quality reports (Galaxy Version 0.72+galaxy1))’ tool with FastQC (Version 0.11.8) [29] were used to generate quality reports. The quality-filtered reads were filtered for plant RNA by mapping trimmed reads onto non-informative host plant sequences using the ‘Map with Bowtie for Illumina (Galaxy Version 1.2.0)’ tool with bowtie (Version 1.2.0) [30]. Filtered reads were subjected to size selection to select the 21–22 nt read fraction using the ‘fastp (fast all-in-one pre-processing for FASTQ files (Galaxy Version 0.20.1+galaxy0))’ tool. For the control dataset, the 24 nt read fraction was also derived, and the downstream workflow was separately conducted on this read fraction. De novo genome assembly was performed with the ‘velveth (Prepare a dataset for the Velvet velvetg Assembler (Galaxy Version 1.2.10.3))’ tool with a kmer length of 15 and the ‘velvetg (Velvet sequence assembler for very short reads (Galaxy Version 1.2.10.2))’ tool, which uses velvet (Version 1.2.10) [31]. The ‘cap3 (Sequence Assembly tool (Galaxy Version 2.0.0))’ with cap3 (Version 10.2011) [32] was used for scaffolding. The open reading frames were predicted using the ‘getorf (Finds and extracts open reading frames (ORFs) (Galaxy Version 5.0.0.1))’ from the EMBOSS suite (Version 5.0.0) [33]. The custom plant virus and viroid database PVirDB was generated within GA using the multiFASTA file as an input for the ‘NCBI BLAST+ makeblastdb (Make BLAST database (Galaxy Version 2.10.1+galaxy0))’ tool with the BLAST (Version 2.10.1) [34]. The scaffolds from cap3 were used as queries for all-vs-all sequence similarity search against the PVirDB database using ‘NCBI BLAST+ blastn (Search nucleotide database with nucleotide query sequence(s) (Galaxy Version 2.10.1+galaxy0))’ tool using BLAST (Version 2.10.1) [34]. The predicted ORFs were used as queries against an all-vs-all sequence similarity search of the translated PVirDB using the ‘NCBI BLAST+ tblastn (Search translated nucleotide database with protein query sequence(s) (Galaxy Version 2.10.1+galaxy0))’ with BLAST (Version 2.10.1) [34]. The scaffolds from the cap3 and ORFs from getorf in FASTA format, and the sequence similarity blastn and tblastn outputs were downloaded for further processing. Summarizing and curation of the blastn outputs were performed using BlastTools java script [35] to obtain a summary output with statistics of top hits for each sample.

#### 2.4.3. GA-VirReport-Stats Workflow

All positive detections were subjected to a secondary complementary workflow, the GA-VirReport-Stats. The 21–22 nt read fraction and the PVirDB database were selected as input files. The accession number of the top hit for an individual detection, which was sourced from the blastn summary output, was also entered manually before executing the workflow. The following section describes the tools used in the GA-VirReport-Stats workflow.

Reference sequences were extracted from the PVirDB database with the ‘NCBI BLAST+ blastdbcmd entry(s) Extract sequence(s) from BLAST database (Galaxy Version 2.10.1+galaxy0)’ tool, which uses BLAST (Version 2.10.1) [34]. The 21–22 nt reads were mapped onto this reference sequence with Map with Bowtie for Illumina (Galaxy Version 1.2.0), which uses bowtie (Version 1.2.0) [30]. ‘SAM-to-BAM convert SAM to BAM (Galaxy Version 2.1.1)’ tool, which applies samtools (Version 1.9) [36], was used to convert the obtained the sequence alignment map to BAM format. Coverage and depth statistics were generated for the mapped data with the ‘QualiMap BamQC (Galaxy Version 2.2.2d+galaxy3)’ tool that uses Qualimap (Version 2.2.2d) [37]. The ‘bcftools mpileup Generate VCF or BCF containing genotype likelihoods for one or multiple alignment (BAM or CRAM) files (Galaxy Version 1.10)’ tool was used to generate VCF/BCF files, and variant calling was performed with the ‘bcftools call SNP/indel variant calling from VCF/BCF (Galaxy Version 1.10)’, which runs bcftools (Version 1.10) [36]. The coverage of the reference sequence/genome was derived with the ‘bedtools Genome Coverage compute the coverage over an entire genome (Galaxy Version 2.30.0)’ [38], which uses BEDTools (Version 2.30.0) [39]. Regions of the reference sequence with no coverage were extracted with the ‘Text reformatting with awk (Galaxy Version 1.1.2)’ tool [38] that uses gawk (Version 4.2.0) and masked with Ns using ‘bedtools MaskFastaBed use intervals to mask sequences from a FASTA file (Galaxy Version 2.30.0)’ tool, which uses BEDTools (Version 2.30.0) [39]. A reference-guided consensus sequence was derived from the VCF variants with ‘bcftools consensus Create consensus sequence by applying VCF variants to a reference FASTA file (Galaxy Version 1.10+galaxy1)’, which uses bcftools (Version 1.10). A final sequence similarity search of the consensus sequence against the PVirDB was performed using ‘NCBI BLAST+ blastn Search nucleotide database with nucleotide query sequence(s) (Galaxy Version 2.10.1+galaxy0)’ tool as described above [34]. The reads per kilobase of transcript per million mapped reads (RPKM) in Equation (1) was computed as follows for each virus and viroid in a sample:(1)$$\mathrm{RPKM}\;=\frac{\mathrm{Number}\;\mathrm{of}\;\mathrm{reads}\;\mathrm{mapped}\;\mathrm{to}\;\mathrm{the}\;\mathrm{reference}\;\times 10^{9}}{\mathrm{Total}\;\mathrm{number}\;\mathrm{of}\;\mathrm{quality}\;\mathrm{filtered}\;\mathrm{reads}\;\times \;\mathrm{reference}\;\mathrm{length}\;\mathrm{in}\;\mathrm{bp}}$$

#### 2.4.4. Curated Plant Virus–Viroid Database: PVirDB

Figure 2 depicts the workflow used to generate PVirDB. All available nucleotide sequences (*n* = 17,153) of regulated viruses and viroids at PEQ and endemic viruses of interest to Australian plant industries were retrieved in GenBank format using an Entrez text search on the NCBI nucleotide database [40]. The obtained list was checked manually, and any non-specific sequences removed. From here on, the above datasets will be referenced as ‘Regulated’ and ‘Endemic’, respectively, and ‘PEQ dataset’ collectively. To prevent off-target detections, we also included in the custom database other reference viral sequences derived for higher plant species and abundant host plant sequences (i.e., chloroplast genomes). A nucleotide accession list of 120,945 viruses from higher plants, Embryophyta (taxid 3193), was obtained from the NCBI Virus [41] (October 2021), and their nucleotide sequences were downloaded in GenBank format using NCBI Batch Entrez [42]. From here on the above dataset will be addressed as ‘Higher plant dataset’. Additionally, complete chloroplast genome sequences from the same genera of host plants held at PEQ were retrieved from NCBI nucleotide database [40] in GenBank format and will be referred to as ‘Chloroplast dataset’ from here on. The GenBank-formatted sequences were converted into FASTA format with annotated taxonomic information using a custom script by Gutiérrez et al. [43]. In the regulated dataset, the unknown base percentages of each sequence were determined using the Seqtk version 1.3 [44], and sequences with >5% N content were removed along with host plant sequences and sequences >20 Kb, except for complete chloroplast genomes that ranged from 100 to 200 kbp. To avoid ambiguous detection and reporting of regulated and endemic pests, we clustered highly similar sequences (≥99.0%) using CD-HIT version 4.8.1 [45] to select a single representative sequence per cluster. Similarly, we clustered the higher plant dataset and retained a single representative sequence for each cluster. Next, duplicated accessions in PEQ and higher plant datasets were removed from the higher plant dataset. Furthermore, highly similar sequences in higher plant dataset to the PEQ dataset (identity > 97% and coverage > 70%) were also removed. The representative sequences for ‘regulated’, ‘endemic’ and ‘higher plant’ datasets were merged into a single FASTA file with chloroplast genomes. To make the database globally compatible, we removed the information on biosecurity nature from the headers of the sequences in the public version of the database.

We compared their sequence similarity to exemplar sequences from the International Committee for Taxonomy of Viruses (ICTV) to validate the selected representative sequences. Where appropriate, we ingested these ICTV exemplar sequences into PVirDB. To this end, we initially obtained virus sequences from the Virus Metadata Repository file available on the ICTV website [46,47]. This file was then filtered, removing all non-exemplar sequences and non-plant host viruses, generating an ICTV plant virus exemplar list. This list was then used as a guide to verify and correct genus and family taxonomic data in the PVirDB fasta headers. ICTV exemplar sequences that were not already represented in the PVirDB within a 97% sequence identity and 95% sequence coverage threshold were appended to the PVirDB FASTA file as new entries. A copy of the dataset for PVirDB was made publicly available at Zenodo’s data repository (https://doi.org/10.5281/zenodo.6609576).

## 3. Results

### 3.1. Pilot Study Implementing and Testing the GA-VirReport Bioinformatics Workflow

In this study, we re-sampled host plant small RNAs of eight virus-positive reference plants held at PEQ (Table 1) from the same pool used in Gauthier et al. [20] sequenced using unique dual indexing adapters on a NovaSeq 6000 sequencing instrument. These known positive control plants from diverse genera, including *Citrus*, *Prunus*, *Iris*, *Ipomoea*, *Miscanthus* and *Rubus*, were used to test the implementation of the GA-VirReport bioinformatics workflow in the GA platform and compare its performance against VirReport and routine PEQ diagnostic assays.

#### 3.1.1. Sequencing and Assembly Statistics

We generated, on average, 48.79 million reads per sample derived from the sequencing of eight positive control plants. Upon removing adapters, deconvolution of UMIs, and filtering of poor-quality bases and reads, we retained an average of 42.49 million (87.11%) high-quality reads per sample (Appendix A). Next, we removed non-informative sequences that mapped onto plant rRNAs, mitochondria and chloroplasts, leaving, on average, 12.39 million high-quality 18–25 nt-long reads per sample. For detecting viruses and viroids, we used the 21–22 nt- or the 24 nt-long read fractions, which were on average 7.36 million (15.09% of the raw reads) and 3.32 million (6.80% of the raw reads), respectively. Only 1.52 million reads were preserved in the 21–22 nt fraction for sample MT002.

#### 3.1.2. GA-VirReport Detected All PEQ-positive Viruses in Control Plants Using the 21–22 nt Read Fraction

All viruses and viroids detected by PEQ assays in the eight control plants were also detected by the GA-VirReport pipeline (Table 1) when using the 21–22 nt read fraction for de novo assembly and utilizing BLASTN to search for sequence similarity to PVirDB and NCBI NT databases (Appendix A). BLASTN searches against both PVirDB and NT generated the same virus/viroid detection for all samples, therefore validating the use of the PVirDB database.

For all cases except MT010, where a novel potyvirus tentatively named *Miscanthus sinensis* mosaic virus (MsiMV) was recently reported [48], the average percentage identities of the contigs to the top hit in the PVirDB database were above 92%. In MT010, the contigs showed on average 76.1% and 85.1% sequence similarity to the maize dwarf mosaic virus (MDMV) and sorghum mosaic virus (SrMV), respectively. The 1× read coverage of mapping 21–22 nt reads onto the respective reference sequence was over 80% in most cases except for the novel *Potyvirus* in MT010 and for citrus vein enation virus (CVEV) in MT003. The number of 21–22 nt reads mapping onto the CVEV genome (MN187043.1) was 835, and the percentage of bases covered at 1× depth was 59.1% (Figure 3 and Appendix A). However, the percentage of bases of the reference covered at 10× depth with 21–22 nt reads was over 70% for most cases, except the prunus necrotic ringspot virus (PNRSV) in MT002 at 35.44%, the novel *Potyvirus* in MT010, ranging from 20.85% to 31.12%, and CVEV in MT003, decreasing to just 7.87%. The percentage coverage of the reference genome by the aligned de novo-assembled contigs, defined as ‘contig coverage’, was also lowest in CVEV (0.82%). Compared to the other viruses in the MT003, with reads per kilobase million (RPKM) values ranging between 2366 and 5804, the lower value 7.40 in CVEV suggested the titer of the virus was significantly low in MT003 (Appendix A). These results suggest that a low number of 21–22 nt reads due to the low titer of the CVEV have impaired de novo assembly, generating only one short contig. The 10× read coverage showed a significant positive correlation (*p* < 0.05, R^2^ = 0.22) to the contig coverage. No significant correlation was observed between the 1× read coverage and contig coverage from 21 to 22 nt reads, suggesting the depth of the reads mapping to the reference is crucial for generating read overlaps necessary for successful de novo assembly.

Identification of predicted ORFs showing homology to viral protein sequences in NCBI NT database provided additional evidence for the presence of infecting viruses detected via BLASTN (Appendix A). Among the ORFs predicted from the contigs assembled using the 21–22 nt fraction, nine ORFS showed above 98.6% sequence identity to citrus tristeza virus (CTV) in MT003, and one ORF showed 99% sequence homology to the raspberry bushy dwarf virus (RBDV) replicase protein (AGH55579.1) in MT005. In MT016, nine ORFs showed sequence homology above 98% to sweet potato feathery mottle virus (SPFMV) polyproteins, and one ORF showed homology to a putative disease resistance protein RGA4 [49].

#### 3.1.3. The Sparser Number of Reads Recovered in the 24 nt Read Fraction Can Impair Viral Detection Relying on De Novo Assembly

The mapping read count and the percentage reference bases covered at 10× were significantly lower (*p* < 0.01) for the 24 nt read fraction than the 21–22 nt read fraction. Although the overall number of contigs generated by the 24 nt pipeline was significantly higher than that generated by the 21–22 nt pipeline (*p* < 0.05) (Appendix A), the number of contigs showing homology to viruses/viroids was significantly higher in the 21–22 nt pipeline (*p* < 0.05). We also observed an increase in the contig coverage of the 21–22 nt read fraction compared to the 24 nt read fraction (*p* < 0.01). The minimum scaffold/contig length returned from both pipelines was 29 bp, whereas the maximum scaffold lengths for the 21–22 nt and 24 nt pipelines were on average 366 bp and 262 bp, respectively. Interestingly, the 24 nt pipeline scaffolds’ mean length was significantly shorter *(p* < 0.01) than that of their 21–22 nt counterparts. These results suggest that the 24 nt pipeline generates a larger number of short contigs during de novo assembly, which cannot be merged into scaffolds due to insufficient read overlap.

The PNRSV in MT002 and CVEV in MT003 were not detected through the de novo assembly of the 24 nt read fraction. Direct mapping onto the tripartite reference genomes of PNRSV aligned significantly more reads from the 21–22 nt read fraction than the 24 nt read fraction (1243 vs. 18 reads onto the PNRSV RNA1 (NC_004362.1); 1443 vs. 8 reads aligned onto the PNRSV RNA2 (NC_004363.1), and 13,437 vs. 107 reads aligned to the PNRSV RNA3 (NC_004364.1) for the 21–22 nt and 24 nt read fractions, respectively (Figure 4). Both read fractions showed the highest coverage across the PNRSV RNA3 with 84.47% of bases of reference covered by 21–22 nt reads and only 2.5% of bases covered by 24 nt reads at 10× read depth. Although the read coverage of the 24 nt reads for the PNRSV RNA3 was significantly higher than the read coverage for the PNRSV RNA1 and PNRSV RNA2, the overlap between reads were not significant to assemble contigs (Figure 4).

Similarly, for sample MT003, alignment of reads onto the reference CVEV genome (MN187043) yielded 835, and 10 mapped reads were derived from the 21–22 nt and 24 nt read fractions, respectively (Figure 3). The genome coverage of CVEV at 10× read depth represented 7.8% and 0% when mapping the 21–22 nt or 24 nt fractions, respectively. A lower mapping depth of 24 nt reads resulted in fewer read overlaps, impairing de novo assembly of both CVEV and PNRSV genomes.

#### 3.1.4. GA-VirReport Detected Additional Viruses in Control Plants

All viruses and viroids detected by PEQ assays and HTS using VirReport [20] were also detected by the GA-VirReport pipeline using both 21–22 nt and 24 nt read fractions (Table 1). A rubus yellow net virus (RYNV), citrus virus A (CiVA) and cherry rasp leaf virus (CRLV) were detected in *Prunus*, *Citrus* and *Iris* control plants, respectively. Two viruses, the sweet potato symptomless virus 1 (SPSMV-1), and sweet potato badnavirus (SPBV), were also found in the sweet potato control plant (Appendix A).

Overall, the average percentage identity of the contigs mapping to the top hit accessions of these viruses was above 91.98%. The genome coverage at 10× was above 49.05%, with the lowest for both statistics reported for the CRLV in MT012. The RPKM value of the CRLV (673.31) was significantly lower than that of the iris severe mosaic virus (ISMV; 1712.50) and tobacco ringspot virus (TRSV; 3786.39) reported from the same sample. Predicted ORFs with homology to known proteins provided additional support for these detections. In MT008, an ORF showed homology (with 96% identity) to the nucleocapsid protein of CiVA (QEE94613.1), and in MT012, three ORFs showing homology to CRLV with 91–97% identities were recovered.

#### 3.1.5. Cross-Sample Contamination Is Not Detected

Gauthier et al. [20] reported the detection of viruses and viroids identified in control plants in other multiplexed samples in the same HTS experiment. In this study, to avoid index-hopping events found in HTS data, we used a different library preparation kit with unique dual indexes. We did not detect any cross-sample contamination across any of the control samples tested. Moreover, none of the flagged false positives previously reported in a subset of control samples used in the current study were detected, thus supporting the cross-sample contamination cases proposed by Gauthier et al. [20]. 

#### 3.1.6. Short Contigs of Plant Material Result in Misassignment in Sequence Similarity Searches against the PVirDB

We found four non-specific detections from the blastn searches of 21–22 nt (*n* = 1) and 24 nt (*n* = 3) read fractions due to the presence of short contigs (20–44 bp) showing high sequence similarity (>87.5%) to accessions in the PVirDB over a short alignment length (<57 bp). Furthermore, those non-specific cases showed low genome coverage at 1× (<11.28%) (Appendix A). To assess if the assembled short contigs originated from host plant sequences [50], we subjected the short contigs to sequence similarity against the NCBI NT database. The top hits were non-coding RNAs (ncRNAs), including mitochondria and U1 spliceosomal ncRNA sequences showing higher nucleotide identities (≥98.21%) than top hits in PVirDB (Appendix A). The exception was a short 30 nt-long contig that showed 100% sequence similarity to both a mitochondrion (LC697740.1) and a citrus exocortis viroid isolate (CEVd; MT561434.1). The use of megablast as the default blastn algorithm prevented the detection of non-specific hits in the 21–22 nt fraction. However, CEVd non-specific hits remained in the 24 nt fraction. Further investigation revealed that the CEVd sequence MT561434.1 was showing top similarity to FJ751930.1, FJ662762.1, DQ318794.1 and DQ318790.1, which are known host-derived sequences erroneously annotated in GenBank as CEVd [51,52]. Therefore, the CEVd sequence set was revised, and sequences with high sequence similarity to plant sequences were removed from PVirDB.

#### 3.1.7. GA-VirReport-Stats Pipeline Produces Longer Consensus Sequences

We implemented a complementary GA-VirReport-Stats pipeline that uses the top BLASTN hits reported by the GA-VirReport pipeline to estimate read mapping statistics for each detected virus and viroid including genome coverage at different read depths, mean genome coverage and normalized read counts (RPKMs). Additionally, the GA-VirReport-Stats pipeline creates a reference-guided consensus viral genome sequence for each positive detection. The initial de novo assembly strategy minimizes false positives, and the subsequent reference-guided approach aims to extend the genome coverage of the detected viruses and viroids. We found, on average, a 52.43% increase in genome coverage by consensus sequences compared to de novo-assembled contigs (Appendix A). Top hits with sequence similarity to de novo-assembled contigs below 90% did not yield complete consensus sequences due to mapping small RNAs with more than three mismatches onto a reference genome, which allows for detecting sequences with up to 10–12% sequence divergence. The exception of a hit with high sequence similarity was CVEV; despite showing 100% sequence similarity to MN187043, the consensus genomic sequence only resolved 59.1% of the reference genome (Appendix A). The low expression titer of CVEV with an RPKM value of 7.40 and only 7.87% base coverage at ≥10× is likely associated with the presence of a large fraction of Ns in the predicted consensus sequence.

### 3.2. Large-Scale Testing and Implementation of GA-VirReport Bioinformatics Workflow on Imported Plants under Quarantine

To assess the detection of plant viruses and viroids in quarantined plants held at the PEQ, we selected 105 plants, including *Prunus spp.* (*n* = 69), *Fragaria ananassa* (*n* = 14), *Pyrus communis* (*n* = 5), *Malus domestica* (*n* = 4), *Solanum tuberosum* (*n* = 9) and *Vitis vinifera* (*n* = 4) (Appendix A). The following sections focus on the results of this large-scale study.

#### 3.2.1. Sequencing Statistics

Small RNA-seq HTS was performed in two batches, batch 1 with 59 plant samples (9 *S. tuberosum* and 50 *Prunus* spp.) and batch 2 with 46 samples (19 *Prunus* spp., 5 *P. communis*, 4 *M. domestica*, 4 *V. vinifera* and 14 *F. ananassa*). Batch 1 and 2 sequencing produced an average of 57 million and 39.2 million raw reads per sample, respectively (Appendix A). Upon quality filtering, the read fraction of 21–22 nt was higher than 3 million for all 17 plant samples with positive detections (Appendix A).

#### 3.2.2. GA-VirReport Detected a Variety of Pests in Plants Quarantined at PEQ

Based on our results from the pilot study on positive control plants, we only used the 21–22 nt read fraction and megablast with the GA-VirReport pipeline to detect viruses and viroids in the subsequent large-scale study of imported plants. Out of 105 plants tested, 14 showed the presence of a total of 23 viruses/viroids. These included 12 *Prunus* spp. (4 nectarines, 2 Japanese plums, 3 sweet cherries, 1 peach, 2 hybrids of apricot and plum), 1 *M. domestica* and 1 *V. vinifera* (Table 2 and Appendix A).

The average nucleotide percentage identity of the contigs to the respective virus/viroid sequence showing the best homology was over 98% for all detections, except for CVA in MT385, which was 92.4% identical to LC523004 (Table 2). The percentage of the reference sequence covered by the alignment of de novo-assembled contigs was highest (77.84%) in the nectarine-stem-pitting-associated virus (NSPaV) detection in MT365. The 1**×** genome coverage for most viruses was ≥87.67%. Additionally, in MT350, one ORF showed 99% sequence identity to the PDV coat protein (ACY25757.1), and another predicted ORF showed 95.18% identity to replicase P1 of PDV (QIQ53097.1).

### 3.3. Curation of a Custom Plant Virus and Viroid Blast Database (PVirDB)

The PVirDB blast database comprises 49,206 individual entries. We verified the taxonomic information against the ICTV plant virus exemplar list, evaluating the latest available taxonomic classification. As ICTV is the accepted authority on virus taxonomic classification, the taxonomic information collected from GenBank records that were misassigned, outdated, or missing was replaced with ICTV taxonomic information. Among the FASTA entries in the PEQ dataset derived from GenBank data, 10.2% had missing genus data, and 20.0% had missing family data. Most entries for viroid species (7.8% of the PEQ dataset) did not contain family or genus information. We also found 222 incorrect genera and 291 erroneous family information data in GenBank records compared to ICTV taxonomic information. Among these entries, there was outdated taxonomic information that was no longer recognized by the ICTV but remained in GenBank files, such as: the *Mandarivirus*, which has since been demoted to a subgenus of *Potexvirus* [53], *Nucleorhabdovirus*, which has been split into three genera, namely *Alphanucleorhabdovirus*, *Betanucleorhabdovirus* and *Gammanucleorhabdovirus* [54], and the family *Luteoviridae*, which has since been abolished [55]. Specific taxonomy changes affecting 22.6% of the PEQ detections are noted in Appendix A. After incorporating the current curations, the PVirDB database contains representative sequences for 47,312 viruses and 1894 viroids belonging to 43 families and 185 genera (Appendix A).

## 4. Discussion

HTS technologies have transformed many fields of life sciences, including pest diagnostics. The ability to detect one or more infecting plant viruses or viroids without a priori knowledge enables reporting of all pests that may be present in a sample using a single assay. However, a critical factor that can increase the complexity of detections is index hopping events between multiplexed samples in the same HTS run, potentially leading to false positives [20]. In this study, we adopted an HTS library preparation protocol that prevents cross-sample contamination, ensures diagnostic accuracy and mitigates the need for post-processing procedures to identify false positives. We also present GA-VirReport, a user-friendly web-based analytical workflow for detecting plant viruses and viroids using small RNA-seq data derived from quarantined plants held at a PEQ.

The automated GA-VirReport workflow runs from processing raw data to detecting viruses and viroids against a custom blast database PVirDB. A custom script runs the BlastTools java script [28], which parses GA-VirReport outputs and summarizes detected pests in each sample. A subsequent GA-VirReport-Stats workflow uses the best hits identified by BlastTools to generate reference-guided consensus genomic sequences and mapping statistics for plants with positive detections. Both GA-VirReport and GA-VirReport-Stats run on the GA platform, an open, web-based analytical environment that enables plant pathologists and molecular biologists with limited bioinformatics expertise to execute diagnostic pipelines. The automated workflows are publicly available as sharable workflows that GA users can import into their profile and use with either the GA-integrated NT database or the PVirDB introduced through this study. The user-friendly one-click solution nature of the workflows, the descriptive statistics and the graphical quality reports reduce the bottleneck for bioinformatic data analysis and interpretation when using HTS in quarantine testing of plant viruses and viroids. Moreover, the reference-guided consensus sequence generated through the GA-VirReport-Stats pipeline enables primer design for independent validation for pathogens where PCR assays are unavailable. The mapping statistics generated through this post-processing pipeline provide an additional level of confidence supporting positive detections.

In this study, we also developed a curated plant virus and viroid database: PVirDB. Identifying viruses and viroids to the lowest taxonomy levels is crucial for regulatory decisions regarding the high-risk plants processed at quarantine facilities [2,3]. Usage of public databases for diagnostics can be challenging due to a lack of consistency among descriptors, incorrect user annotations and the large scale of public databases [56,57]. PVirDB contains a curated subset of representative sequences of viruses reported from higher plants in NCBI. With harmonized sequence descriptors with taxonomy (Genus, Family) information curated with the latest ICTV taxonomy release, PVirDB provides accurate taxonomic assignment of viruses and viroids detected through the GA-VirReport workflows. The portability of the database makes it compatible with other wet-lab protocols and bioinformatics pipelines for plant virus and viroid detection as well as other applications such as virus discovery and phylogenetics. Sequences in public databases can have erroneously assigned species names. For example, a plant-derived sequence erroneously annotated as CEVd caused false positives in the analysis of control plants. Additional checkpoints, including scrutinizing the sequence similarity of representative virus/ viroids sequences against ICTV exemplar sequences based on species demarcation criteria, will ensure the identification of other misannotated entries.

We detected viruses and viroids from 22 of the 113 plants tested through the GA-VirReport pipeline. We recovered all true-positive viruses and viroids in the plants diagnosed by PEQ assays [20]. The fact that the true-positive detections were the same across different sequencing platforms, commercial kits, sequencing providers and different sampling times indicates the robustness of the HTS as an approach for the diagnosis of plant viruses and viroids.

The abundance of pests varied among samples, with mapping read counts ranging from hundreds to thousands. However, in the control samples MT003 and MT008, where more than one virus was present, the viral titer did not necessarily match the abundance reported previously. In MT003, Gauthier et al. [20] reported CDVd as the lowest abundant pathogen, but in this study, it was CVEV. In the same study, HSVd was the lowest abundant pathogen in MT008. However, in this study, it was CDVd. The changes observed in titer among different viruses in the same plant sampled a year apart may highlight temporal differences. The viral titer within a host is primarily affected by seasonality [58]. An increase in environmental temperature alters the competition among viral strains within the plant hosts [59]. However, Gauthier et al. [20] and this study maintained the tested plants in PEQ glasshouses under the same growing conditions. Thus, changes in environmental factors are unlikely to explain the observed virus titer differences. Another possibility is the change in tissue sampling location between studies. Many viral determinants facilitate cell-to-cell and long-distance movement of the virus, eventually causing the infection to become systemic [60,61]. However, it is also well-documented that different viruses vary in their potential to cause systemic disease, and viral titer can be affected by the change in sampling location. Additionally, using different cDNA library kits can yield different biases [46], potentially affecting the small RNA content in samples. Finally, the interaction between co-infecting viruses and the plant immune response could impact viral abundance over time. These observations emphasize the importance of cDNA library kit selection, sampling time, growing conditions and collecting multiple tissue types for viral diagnostics.

We did not observe cross-sample contamination among samples tested in the pilot and large-scale studies presented here. Although this can occur at various stages during the wet-lab procedure, contamination due to index misassignment is an inherent issue of HTS using Illumina’s ExAmp patterned flow cells [62]. Index misassignment occurs in pooled libraries due to the residual excessive primers and adapters used in library preparation, causing spurious extension of fragments with the incorrect sample index. When single or combinatorial dual indexing is used in pooled libraries, swapped indices can cause misassignment of samples [63]. This phenomenon is especially problematic in quarantine testing, as this can lead to false positives and potential destruction of a healthy plant if no confirmatory independent molecular assays are performed. Processing a single sample in an HTS run is not a cost-effective strategy to mitigate index swapping events when processing a large volume of samples. Gauthier at al. [20] reported cross-sample contamination events in two different sRNA-seq approaches that used DNA nanoball (DNB)-based HTS and the Illumina NextSeq 500 platform, which are reported to have low levels of index misassignment compared to patterned flow cells [64]. Usage of unique dual indexing, blocking adapters and pooling libraries together just before sequencing has been proposed to mitigate cross-sample contamination [62,63]. When unique dual indices (UDIs) are incorporated during library preparation, misassigned reads can be identified via demultiplexing using both i7 and i5 indexes [63]. The use of unique molecular identifiers (UMIs) in addition to the i7 and i5 adapters can further increase sensitivity towards low-frequency variant detection and reduce PCR-induced errors in library preparation [64]. In the current study, we incorporated UDIs together with UMIs through the QIAseq miRNA UDI 96 index plate, which yielded accurate assignment of sequencing reads across all pooled libraries. Moreover, libraries were cleansed using a bead-based approach to remove residual primers and adapters [62,64]. These findings reinforce the necessity of using UDIs and optionally UMIs for small RNA-seq datasets as standard best practices to ensure the accuracy of HTS-based quarantine testing, especially with sequencing instruments that use patterned flow cells such as NovaSeq.

The host antiviral response against viruses is driven by Dicer 4 (DCL4) and Dicer 2 (DCL2) enzymes that lead to the cleavage of viral dsRNA molecules into small RNAs 21nt and 22 nt in length, respectively [65,66]. Both DCL4 and DCL2 are essential for host plant viral defense by RNA silencing [66]. Additionally, 24 nt-long sRNAs are recognized and cleaved by Dicer 3 (DCL3), an enzyme involved in the silencing of transposons and repetitive elements [67] that has also been reported to be involved in immunity against DNA viruses by inducing methylation in viral cytosines [68,69,70]. Interestingly, DCL4 can act cooperatively with DCL3 in triggering the antiviral response of DNA viruses [70]. Barrero et al. [10] reported that the 21–22 nt sRNA fraction detected most viruses, except for a citrus endogenous pararetrovirus that was better assembled with the 24 nt read fraction. Therefore, we initially implemented two separate assembly pipelines for both size fractions in the pilot study, the 21–22 nt pipeline to detect active viral infections and the 24 nt pipeline to flag possible host-genome-integrated viruses. Our results suggest that using the 21–22 nt read fraction improves and is sufficient for the reliable detection of viruses and viroids by generating longer contigs and reducing the risk of spurious detections. Comparatively shorter contigs generated through the 24 nt read fraction caused false positives due to random hits of assembled host RNA sequences to viral genomes. We detected four DNA viruses, namely, PGVA and SPSMV-1, which are single-stranded, and RYNV and SPBV, which are double-stranded DNA viruses using the 21–22 nt fraction. In all four DNA viruses, we found a greater number of 21–22 nt mapped reads than 24 nt reads, further supporting using the 21–22 nt read fraction for downstream de novo assembly and detection of viral pathogens. Therefore, we have made the 24 nt pipeline optional in the GA-VirReport workflow as the focus of routine diagnostics is to detect active de novo viral infections triggering the host plant RNAi response.

Another important finding was that the megablast algorithm outperformed the blast algorithm in blastn sequence similarity searches performed in this study. Blastn resulted in a higher number of spurious hits to the sequences in the PVirDB compared to the megablast in the pilot study. Compared to the blast algorithm, megablast is more optimized for finding closely related sequences with percentage identities >95%. Megablast can handle longer DNA sequences, and due to the word size being larger, it is also faster than blastn [71]. In the context of screening for regulated pests where we are interested in finding the best match to known pests rather than new-to-science pests, megablast was therefore the favored option.

While our bioinformatics toolkit provided the same level of sensitivity compared to existing PEQ molecular assays and reported additional pathogens that are not routinely tested at PEQ, an integrated method for distinguishing false negatives/positives and true negatives/positives could provide an additional layer of confidence. This is of paramount importance in HTS approaches, including in this study, where the number of total reads generated by the sequencing instrument did not necessarily correlate to the ‘number of usable reads’, defined in the current study as the number of remaining reads after filtering out plant ribosomal, mitochondrial and chloroplast RNAs. Usable reads are 21–22 nt-long and can be used for de novo assembly. Inadequate recovery of ‘usable reads’ can cause false negatives if the viral titer is low. The number of usable reads is affected by the sampling methods, efficiency of the RNA extraction from different plant commodities and the success of library preparation, which may be prone to variability. Although sensitivity can be improved by increasing the sequencing depth [17], we have observed that attempts at deeper sequencing as high as 75 million reads to compensate for the low titer in the virus or poor quality of the RNA sample tends to introduce false positives. Massart et al. [17] proposed a minimum sequencing depth of 2.5 million raw reads for sRNA viral diagnostics, and Gauthier et al. [20] reported that at least 2.5 million quality-filtered reads were required to obtain 100% sensitivity. In the current study, the minimum number of 21–22 nt reads that was used to detect a pathogen was 1.51 million reads. However, we have observed that defining a universal threshold for the ‘usable reads’ is challenging due to variability in the quality and composition of RNA from different plants and in the genome size and titer of the pathogens present.

The use of appropriate controls at different sample preparation and sequencing steps is important to generate high-quality HTS data. Although controls were used to assess the library preparation and sequencing success by the sequencing service provider, additional controls are recommended to test for efficiency of the entire wet-lab process [72] and to account for variability among commodities. The introduction of an external positive control for each targeted virus introduces a risk of plant contamination at the PEQ [9]. Therefore, usage of known nucleic acids as an internal control is recommended. Kesanakurti et al. spiked plant tissue with leaf discs from a plant with two known endogenous viruses as an alien internal control for the nucleic acid extraction [73]. Allocating additional resources for maintenance of plants may not be cost-effective in post-entry quarantine facilities. Therefore, we propose the use of synthetic oligonucleotides, for example, invertebrate miRNA genes that are not present in plant genomes, as spiked in controls added during the RNA extraction as an internal control for future HTS testing at the PEQ. Detection of a known concentration of internal controls in the sequencing data will enable benchmarking the analytical sensitivity of the HTS approach and enable normalizing the read counts of the pathogens to eliminate noise due to contamination in addition to assuring the success of the wet-lab procedure.

## 5. Conclusions

Small RNA-Seq HTS using UDIs and subsequent processing of 21–22 nt-long read fractions through the GA-VirReport pipeline detected all viruses and viroids reported by PEQ assays and additional viruses reported by the VirReport pipeline. In this study, using a 21–22 nt read fraction that reflects the host antiviral RNAi response was better-suited for detecting plant viruses and viroids. Furthermore, the screening of 105 quarantined plants to assess their phytosanitary status represents a case study for adopting HTS for quarantine testing. HTS, as a first-pass screening method, enables targeted confirmatory testing using molecular assays in HTS-positive detections, thereby accelerating the quarantine testing process.

## Figures and Tables

**Figure 1 viruses-14-01480-f001:**
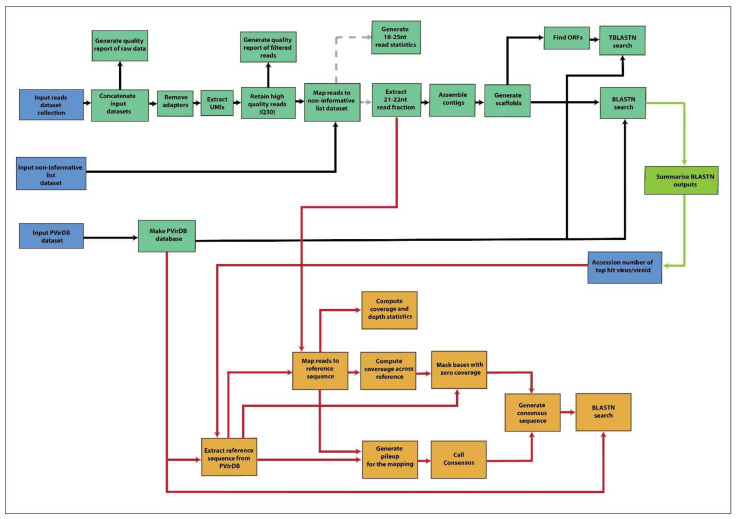
GA-VirReport and GA-VirReport-Stats workflows. GA-VirReport workflow is shown in green, and the GA-VirReport-Stats workflow is shown in orange.

**Figure 2 viruses-14-01480-f002:**
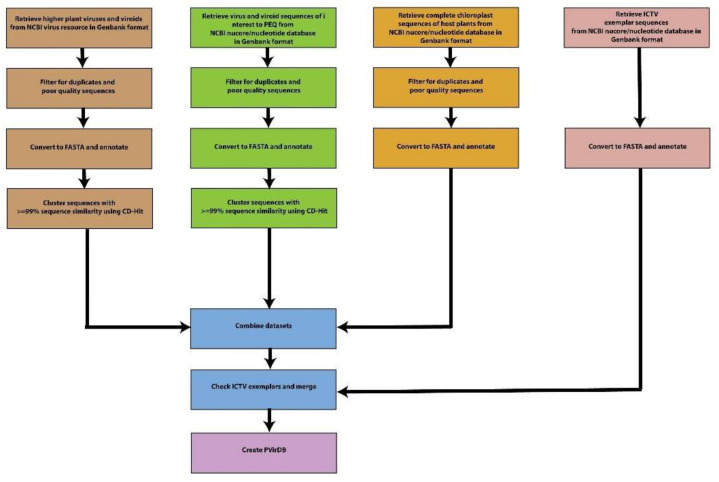
Workflow for generation of PVirDB.

**Figure 3 viruses-14-01480-f003:**
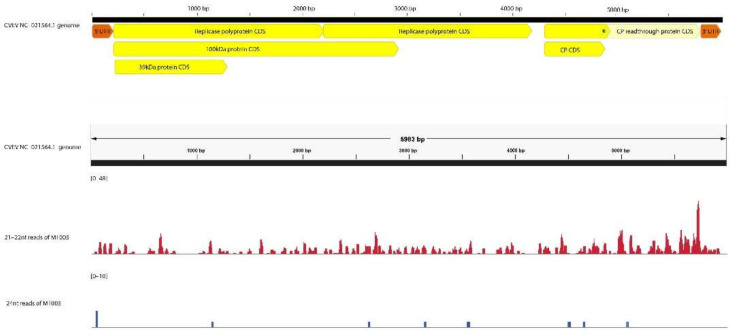
Read coverage of 21–22 nt and 24 nt reads mapping to the citrus vein enation virus genome (NC_021564.1) computed through the GA-VirReport-Stats pipeline. The scales of the mapped read depth for 21–22 nt reads and 24 nt reads are [0–48] and [0–10], respectively.

**Figure 4 viruses-14-01480-f004:**
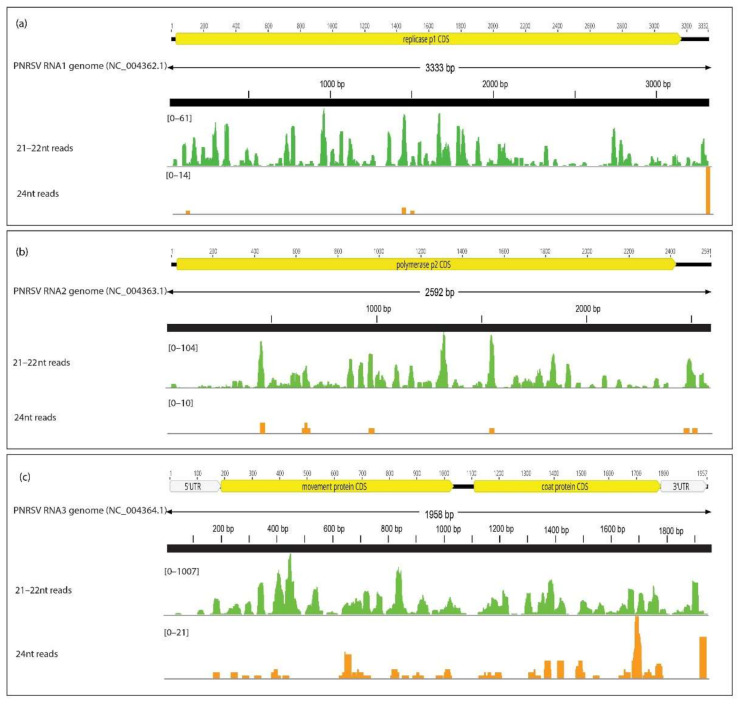
Mapping of filtered 21–22 nt reads and 24 nt reads of MT002 to the tripartite reference genomes of prunus necrotic ringspot virus (**a**) RNA1, (**b**) RNA2 and (**c**) RNA3.

**Table 1 viruses-14-01480-t001:** PEQ-positive control plants used in the pilot study infected with known viruses and viroids.

Sample	Commodity	Species	Known PEQ Detections ^1^	Detections from VirReport ^2^	Detections from GA-VirReport ^3^
MT001	Citrus	Citrus Troyer × Frost-Lisbon	CEVd	CEVd	CEVd
MT002	Stonefruit	*Prunus persica* (Nectarine)	PNRSV	PNRSV	PNRSV
MT003	Citrus	*Citrus aurantiifolia* (Christm.) Swingle	CTV, CVEV, CDVd, HSVd	CTV, CVEV, CDVd, HSVd	CTV, CVEV, CDVd, HSVd
MT005	Raspberry	*Rubus idaeus*	RBDV	RBDV, RYNV	RBDV, RYNV
MT008	Citrus	*Citrus sinensis*	CDVd, HSVd	CDVd, HSVd, CiVA	CDVd, HSVd, CiVA
MT010	Ornamental grass	*Miscanthus sinensis* ‘Morning light’	Novel *Potyvirus* (MsiMV)	Novel *Potyvirus* (MsiMV)	Novel *Potyvirus* (MsiMV)
MT012	Iris	*Iris* sp. ‘Crimson colossus’	ISMV, TRSV	ISMV, TRSV, CRLV	ISMV, TRSV, CRLV
MT016	Sweet Potato	*Ipomoea batatas* ‘GRF0334’	SPFMV	SPFMV, SPSMV-1, SPBV	SPFMV, SPSMV-1, SPBV

^1^ Presence of regulated viruses confirmed using molecular (PCR and ELISA) and bioassays (biological and woody indexing): ^2^ Viruses and viroids reported through the VirReport pipeline according to Gauthier et al. [20]: ^3^ Viruses and viroids reported through GA-VirReport pipeline presented in this study; CEVd = citrus exocortis viroid, PNRSV = prunus necrotic ringspot virus, CTV = citrus tristeza virus, CVEV = citrus vein enation virus, CDVd = citrus dwarfing viroid, HSVd = hop stunt viroid, RBDV = raspberry bushy dwarf virus, RYNV = rubus yellow net virus, CiVA = citrus virus A, ISMV = iris severe mosaic virus, TRSV = tobacco ringspot virus, CRLV = cherry rasp leaf virus, SPFMV = sweet potato feathery mottle virus, SPSMV-1 = sweet potato symptomless virus 1, SPBV = sweet potato badnavirus.

**Table 2 viruses-14-01480-t002:** Detections of viral sequences in the large-scale high-throughput sequencing testing of quarantined plants using GA-VriReport.

Sample ID	Commodity	Species	^1^ GA-VirReport Detection	^2^ Subject Accession	^3^ Average Percent Identity	^4^ Percentage Contig Coverage	^5^ Percentage Bases 10× Read Depth
MT338	Stone fruit	*Prunus salicina*	PrLV	MF510412	100	41.26	85.54%
MT341	Stone fruit	*Prunus salicina*	PrLV	MF510412	100	25.95	66.43%
MT350	Stone fruit	*Prunus avium*	CVA	LC422952	99.41	32.41	79.78%
			PDV	MT013233	98.8	64.01	96.00%
MT357	Stone fruit	*Prunus salicina* × armeniaca	NeVM	KT273412	99.48	47.71	95.01%
MT362	Stone fruit	*Prunus persica* var. nucipersica	NSPaV	KT273410	99.15	76.86	98.24%
MT363	Stone fruit	*Prunus persica* var. nucipersica	NSPaV	KT273410	99.06	76.32	97.44%
MT365	Stone fruit	*Prunus persica* var. nucipersica	NeVM	KT273412	99.16	60.24	96.51%
			NSPaV	KT273410	99.29	77.84	98.16%
MT368	Stone fruit	*Prunus persica* var. nucipersica	NeVM	KT273412	98.61	50.19	96.83%
MT370	Stone fruit	*Prunus persica*	NSPaV	KT273410	99.34	68.48	96.31%
MT378	Stone fruit	*Prunus salicinia* × *avium*	NeVM	KT273412	99.03	56.28	97.82%
MT383	Stone fruit	*Prunus avium*	CVA	KX370827	98.19	38.27	95.67%
			PDV	MT013233	98.81	74.61	97.96%
MT385	Stone fruit	*Prunus avium*	CVA	LC523004	92.44	14.1	98.07%
			PDV	MT013233	98.75	72.99	79.58%
MT410	Apple	*Malus domestica*	CCGaV	MK940543	99.79	33.56	76.70%
			ASPV	KF321967	96.87	24.55	59.02%
			ACLSV	KR606325	98.04	47.2	97.31%
MT411	Grapevine	*Vitis vinifera*	HSVd	MF979532	100	51.34	100.00%
			GRSPaV	MG938345	98.19	43.42	90.52%
			GYSVd	DQ377130	100	34.9	96.12%
			AGVd	GU327604	100	14.68	79.82%

^1^ Viruses and viroids reported through GA-VirReport pipeline presented in this study: PrLV = prunus latent virus, CVA = cherry virus A, PDV = prune dwarf virus, NeVM = nectarine virus M, NSPaV = nectarine-stem-pitting-associated virus, CCGaV = citrus-concave-gum-associated virus, ASPV = apple stem pitting virus, ACLSV = apple chlorotic leaf spot virus, HSVd = hop stunt viroid, GRSPaV = grapevine-rupestris-stem-pitting-associated virus, GYSVd = grapevine yellow speckle viroid, AGVd = Australian grapevine viroid; ^2^ subject accession from PVirDB; ^3^ Average percentage identity of the cumulative alignment and the ^4^ coverage of the subject achieved by the alignment of de novo contig was computed through BlasTools java script [35] summarizing BLASTN outputs of GA-VirReport. ^5^ Percentage bases of the reference covered by 10× mapping depth of filtered 21–22 nt reads.

## Data Availability

The data presented in this pilot study are openly available in the Short Read Archive (SRA) database under the BioProject PRJNA752836 (https://dataview.ncbi.nlm.nih.gov/BioProject/PRJNA752836, accessed on 15 February 2022). The data presented in the large-scale quarantine testing study are not publicly available due to being commercial in confidence client data. Access to viruses’ data can be provided upon request. GA-VirReport workflow for single-input files (https://doi.org/10.5281/zenodo.6387498), multiple-input files (https://doi.org/10.5281/zenodo.6387492), GA-VirReport-Stats workflow (https://doi.org/10.5281/zenodo.6387504) and PVirDB dataset (https://doi.org/10.5281/zenodo.6545171) are available through the Zenodo data repository (https://zenodo.org/, accessed on 27 June 2022).

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
