# Peer review of "Implementation of GA-VirReport, a Web-Based Bioinformatics Toolkit for Post-Entry Quarantine Screening of Virus and Viroids in Plants"

_viruses, 2022, doi:10.3390/v14071480_

Round 1
Reviewer 1 Report
The manuscript by Lelwala et al., introduces GA-VirReport, a novel web platform to perform bioinformatics analysis of small RNA High-throughput sequencings for quarantine virus and viroid identification in plants. Although there are several platforms, either web- or bench- based, already described in the literature that conduct similar analyses, GAVIr focuses on usability and does not require advanced technical knowledge from the user and being based on an online server does not require a powerful computer in the hands of the user. Moreover, the platform uses a curated database of viruses and viroids instead of accessing the entire GenBank or similar databases, speeding up BLAST analyses. Further, the toolkit GA-VirReport has been validated at scale using a large number of samples.
Comments
I do not see in the Material and Methods or Results information about the kmer value(s) used in Velvet for contig generation. It is important to highlight this information in order to be able to compare the contigs obtained at different kmer values and in turn with results obtained on other platforms.
Author Response
Response. We appreciate the feedback provided by the Reviewer. In agreement with the Reviewer's comment, we have updated the methods section as follows:
“De novo genome assembly was performed with the ‘velveth (Prepare a dataset for the Velvet velvetg Assembler (Galaxy Version 1.2.10.3))’ tool with a kmer length of 15 and the ‘velvetg (Velvet sequence assembler for very short reads (Galaxy Version 1.2.10.2))’ tool, which uses velvet (Version 1.2.10) [30]” (Page 6, lines 217-221).
Reviewer 2 Report
The authors present a thorough and detailed study of their work developing new bioinformatics resources for detecting plant viruses and a demonstration of its usability for screening plant material in Australian post-entry quarantine. This study builds on the authors previous study (Gauthier et al 2022) comparing HTS and bioinformatics approaches for PEQ diagnostics. I believe this present manuscript provides a valuable additional contribution in creating an optimised pipeline executable in the Galaxy Australia platform and a curated plant virus database for improved sequence identification.
My main comment is that I’d like to see more information in the ms for how accessible and useful these tools are outside Australia. It’s not clear to me if the GA platform is accessible globally. The PVirDB is also developed for Australian plant industries, but could it be used/adapted for other countries. Some additional/revised statements would help clarify if these tools are solely applicable to Australian users or can be implemented more broadly. Also while the PVirDB is presumably a ‘cleaner’ database than an uncurated NCBI database, how will this be maintained/updated?
Author Response
We appreciate the feedback provided by the Reviewer. In the attached cover letter, we address each of the points raised by the Reviewer.

Reviewer 3 Report
The current study by Lelwala et al. describes a web-based tool ‘GA-VirReport’ that is executable on the Galaxy platform. In my opinion, GA-VirReport is an excellent web-based analytical tool that will help in detecting plant viruses in quarantine samples and in the regular screening of plant viruses in field studies worldwide. Lelwala et al also did an excellent job of curating the representative sequences of viruses reported from higher plants in NCBI. Furthermore, the manuscript is very well written, and I have very minor edits and comments in the attached Pdf.

Author Response
We appreciate the feedback provided by the Reviewer. In the attached cover letter, we provide responses to each of the items raised by the Reviewer.
